# Smartphones and Threshold-Based Monitoring Methods Effectively Detect Falls Remotely: A Systematic Review

**DOI:** 10.3390/s23031323

**Published:** 2023-01-24

**Authors:** Ricardo A. Torres-Guzman, Margaret R. Paulson, Francisco R. Avila, Karla Maita, John P. Garcia, Antonio J. Forte, Michael J. Maniaci

**Affiliations:** 1Division of Plastic Surgery, Mayo Clinic, 4500 San Pablo Rd, Jacksonville, FL 32224, USA; 2Division of Hospital Internal Medicine, Mayo Clinic Health Systems, 1221 Whipple St., Eau Claire, WI 54703, USA; 3Division of Hospital Internal Medicine, Mayo Clinic, 4500 San Pablo Rd, Jacksonville, FL 32224, USA

**Keywords:** remote sensing technology, smartphone, mobile applications, artificial intelligence, hospital-at-home

## Abstract

In the US, at least one fall occurs in at least 28.7% of community-dwelling seniors 65 and older each year. Falls had medical costs of USD 51 billion in 2015 and are projected to reach USD 100 billion by 2030. This review aims to discuss the extent of smartphone (SP) usage in fall detection and prevention across a range of care settings. A computerized search was conducted on six electronic databases to investigate the use of remote sensing technology, wireless technology, and other related MeSH terms for detecting and preventing falls. After applying inclusion and exclusion criteria, 44 studies were included. Most of the studies targeted detecting falls, two focused on detecting and preventing falls, and one only looked at preventing falls. Accelerometers were employed in all the experiments for the detection and/or prevention of falls. The most frequent course of action following a fall event was an alarm to the guardian. Numerous studies investigated in this research used accelerometer data analysis, machine learning, and data from previous falls to devise a boundary and increase detection accuracy. SP was found to have potential as a fall detection system but is not widely implemented. Technology-based applications are being developed to protect at-risk individuals from falls, with the objective of providing more effective and efficient interventions than traditional means. Successful healthcare technology implementation requires cooperation between engineers, clinicians, and administrators.

## 1. Introduction

Falls are one of the most concerning areas in healthcare, and they are frequently associated with devastating consequences such as fractures or neurologic injuries, particularly in older adults. Falls are the most common cause of fatal injuries [1]. In the United States, approximately 28.7% of community-dwelling older adults 65 and older fall at least once per year [1]. A total of 37.5% of those who fell reported at least one fall that required medical attention or limited their activity for at least one day, resulting in an estimated 7.0 million fall injuries [1].

Furthermore, falls are the leading cause of fatal injuries [1] and the most common reason for trauma-related hospitalizations in older adults [2]. Older adults who have fallen in the last two years are two to three times more likely to fall again within a year [1]. Nonfatal injuries secondary to falls are one of the top 20 most expensive conditions, with a medical cost of USD 51 billion in 2015 [3,4], which is expected to rise to USD 100 billion by 2030 [5].

Approximately 25% of hospital falls result in injury, lengthening a patient’s stay, increasing healthcare costs, and increasing liability [6,7,8,9]. Additionally, the Centers for Medicare and Medicaid Services will not reimburse hospitals for care when patients sustain specific fall-related injuries, putting hospitals under significant financial pressure to prevent falls [10].

Several approaches have been implemented to prevent falls in hospitals. This wide range of fall prevention practices includes patient monitoring tools (e.g., sitters), bed modifications (e.g., alarms), identification practices (e.g., bracelets), safety practices (e.g., clutter-free floors), and patient and family education [10].

However, there is scant [11,12,13,14,15,16,17]—and occasionally even contradictory—support for implementing any of these strategies [18,19]. Even though bed alarms, for instance, are ineffective at preventing falls and detrimental (such as noise and alarm fatigue), they are nevertheless often employed in medical facilities [14,18,20,21,22]. The case for using multicomponent interventions is more significant, although it is not apparent which elements have the most important influence on falls [10]. For the prevention of hospital falls, experts advise adopting multicomponent treatments and customizing the procedures for the patients who will be cared for in the unit [10].

There are no accepted, evidence-based therapies for fall prevention, in contrast to other healthcare-acquired illnesses [11,12,13,14,15,16,17,23,24], making it challenging for hospitals to determine which preventative strategies have the most influence on fall rates. This review intends to discuss the scope of smartphone (SP) utilization in fall detection and prevention in various care settings. The review’s focus will be on the applicability, current utilization status, effectiveness, and pros and cons of using SP and threshold-based algorithms to detect and predict falls and future directions in this field.

## 2. Material and Methods

### 2.1. Eligibility Criteria

Studies that described setting thresholds to the SP sensors to detect and/or prevent falls remotely were included. We classified fall detection studies as those that described devices or systems that used an accelerometer or other sensors to detect changes in movement that could indicate a fall and send an alert to a caregiver or emergency contact. Fall prevention studies, on the other hand, are those that describe devices or systems that monitor a person’s movements and provide feedback or suggestions to help prevent falls by using sensors that detect changes in movement and alert the user to any potential fall risk. They may also make suggestions or make recommendations to assist users in adjusting their movements and avoiding falls. Studies that used telehealth video calls or digital exercise programs, studies in languages other than English, reviews, and book chapters were excluded (Figure 1).

### 2.2. Information Sources and Search Strategy

A computerized search was conducted on 12 November 2022, by two independent investigators (R.T.G. and F.R.A.) using the following electronic databases: PubMed (1994–present); MEDLINE (1996–present); Embase (1988–present); CINAHL (1994–present); Web of Science (1900–present); NIH ClinicalTrials.gov (1997–present); Google Scholar. The following MeSH terms were used: “Remote Sensing Technology”, “Technology”, “Wireless Technology”, “Accidental Falls”, Fall detection”, “Accidents, Home”, and “Home”.

### 2.3. Study Selection and Data Collection Process

Two investigators (R.T.G. and F.R.A.) independently conducted the search and filtration following the inclusion and exclusion criteria described above. The studies were filtered based on titles consecutively by abstracts, and lastly by full-text reading. If both authors considered that the article met all criteria, the manuscript was included. In case no consensus could be made, a third author (A.J.F.) designated the inclusion or exclusion of the article. Data from the selected papers are summarized in Table 1.

### 2.4. Risk Bias Assessment

The bias risks of selected studies were assessed with the help of the ROBINS-I tool of the Cochrane Library for nonrandomized studies. A description of individualized bias and cross bias is shown in Figure 2 and Figure 3, respectively.

## 3. Results

### 3.1. Study Characteristics

A total of 44 studies were included following our inclusion and exclusion filtration of manuscripts. Characteristics of the included studies can be found in Table 1. Out of these studies, forty-three aimed to detect falls, two intended to detect and prevent falls, and only one aimed to prevent falls. The totality of the studies used accelerometers as the method to detect and/or prevent falls (Figure 4). Six manuscripts used the smartphone and external sensors during their measurements. Alarm to guardian was the most prevalent course of action following a fall event.

### 3.2. Fall Prevention Studies

Building on Dai et al.’s [25] work in 2010, Fontecha J.N. et al. [26] in 2013 described a mobile system for detecting and diagnosing frailty in healthcare settings. This system incorporated an accelerometer-enabled smartphone (SP) to collect movement data for two basic gait and balance tests, as an extension of Dai et al.’s PerFallId platform for pervasive fall detection using mobile phones.

Mellone S. et al. [27] presented in 2012 a fall detection application for SP use named uFALL. This application can effectively transform an SP into a long-term monitoring device with real-time fall detection capability. Furthermore, this same group of investigators developed a fall prevention app called uTUG, which transforms an SP into a pocket-sized mobility laboratory, allowing for quick screening, assessment, and follow-up. All of the SP-based solutions presented by these researchers are capable of supporting clinical research and practice in fall detection and prevention at various levels in a cost-effective manner (Figure 5).

### 3.3. Fall Detection Studies

Several groups have conducted research into fall detection [25,27,28,29,30,31,32,33,34,35,36,37,38,39,40,41,42,43,44,45,46,47,48,49,50,51,52,53,54,55,56,57,58,59,60,61,62,63,64,65,66,67,68,69]. Yavuz et al. [66] proposed two alternatives in 2010 to the then-current trending model for accelerometer signals with varying frequency content over time in order to differentiate falling activity from other activities. These two alternatives were the short-time Fourier transform (STFT) and wavelet transformation, with the Meyer wavelet proving to be able to distinguish between falls and nonfalls with an 85% recall and 95% precision. The Fourier transform is a mathematical transform that decomposes functions into frequency components, represented as a frequency function by the transform’s output. Most commonly, time or space functions are transformed, producing a function based on temporal or spatial frequency.

Following this, Viet et al. [64] developed a fall detection method using a one-class SVM algorithm in 2011, and Cao et al. [39] presented a fall detection system for Android smartphones in 2012, while He Y. et al. [48] reported a system that uses an SP mounted on the waist to classify human movements in real time. Viet, V. Q. et al. [64] proposed an algorithm to detect falls using a popular smartphone’s accelerometer and orientation sensor in that same year. Boehner et al. [38] proposed a new method for detecting activity patterns using a smartwatch and smartphone connected via Bluetooth. Subsequently, Koshmak et al. [50] proposed a framework for detecting falls using mobile phone technology and physiological data monitoring, and Mehner et al. [55] developed an application to reduce smartphone battery consumption. Aguiar et al. [37] developed an algorithm to detect falls using accelerometer data from a smartphone, and Colon L. et al. [42] tested the accuracy of a sensing device (smartphone) located on the user’s body. Lastly, Maglogiannis et al. [54] created a fall detection app for smartwatches.

In 2011, Lee R. Y et al. [34] identified that the specificity and sensitivity of detecting a fall event using a Smartphone (SP) were high, prompting Lopes I. C. et al. [35] to develop an SP application named SensorFall with the main goal of detecting and notifying falls. This application could accurately detect acceleration values, distinguishing between a fall and false positives and negatives. Suh et al. [36] then studied a congestive heart failure (CHF) remote patient monitoring system (WANDA), which included an SP version that was capable of detecting falls and sending alert messages to caregivers. Following this, Bai et al. [28] in 2012 used three characteristics of the various patterns of SP acceleration values to detect a user’s falling motion while also using the GPS module to determine the fall’s location, expediting the arrival of assistance. Later, Lee J. V. et al. [33] designed and reported a smart elderly home monitoring system (SEHMS) that featured an Android-based SP with a three-axial accelerometer, connected to the system via Wi-Fi. In 2014, Castillo J. C. et al. [29] developed a solution for activity monitoring and fall detection, while Kwolek et al. [32] demonstrated how to improve fall detection using depth and accelerometric data. Finally, Hsieh et al. [31] reported the first study to show that an SP can measure postural stability and distinguish between older adults at low and high risk of falling, showing that SP technology has the potential to improve balance screening in older adults.

Yi et al. [67] proposed a design that uses multiple accelerometers to identify body posture and detect falls in real-time, which could be incorporated into a larger wireless body sensor network for continuous monitoring and providing immediate medical attention. Casilari et al. [40] conducted a system evaluation for two detection devices (smartphone/smartwatch) and found that combining them improved the system’s accuracy in avoiding false positives and false negatives. Madansingh et al. [53] designed a smartphone-based fall detection system and tested its efficacy for daily living activities. Kinematic movement analysis using sensors found in smartphones was used for continuous monitoring, with no false positives. These findings are important for creating machine learning algorithms to reduce false positives and negatives in fall detection.

Pierleoni et al. [57] had difficulty distinguishing between a fall event and a collapse on an armchair in their study, so Vilarinho et al. [65] developed a system with both threshold-based and pattern recognition techniques to help with specificity. Casilari et al. [41] 2016 demonstrated a smartphone-based architecture for automatic fall detection that included a collection of small sensing devices and found that accuracy increased with the number of sensing devices. Figueiredo et al. [44] analyzed smartphone sensors for their ability to distinguish between falls and daily life activities and concluded that the accelerometer was the most reliable sensor. Qu et al. [58] created a system that utilized a monitoring time to detect dangerous falls, built on the Android platform and designed for low energy consumption and fast processing. Tran et al. [61] implemented a fall detection system that analyzed acceleration patterns and an additional long lie detection algorithm to improve the fall detection rate while maintaining an acceptable false-positive rate. Yildirim et al. [68] also created a system to detect dangerous falls using a monitoring time, built on the Android platform and optimized for low energy consumption and fast processing.

Hakim et al. [45] developed a fall detection system using smartphones and standard machine learning algorithms, such as support vector machines, to achieve near-perfect accuracy. He, J. et al. [47] used a wearable motion sensor and a smartphone with two types of algorithms—sliding window and Bayes network classifier—to mimic the fall detection system. Islam Z. Z. et al. [49] created a system that stores accelerometer data generated before and after a fall to detect patterns in accelerometer data prior to falling. The proposed systems take advantage of wearable devices and smartphones as they can detect falls with accuracy and provide timely help for the elderly.

Building upon the research of Tran H. et al. [60], who encountered numerous problems using an analytical method in conjunction with machine learning techniques to distinguish between fall events and common activities, and obtaining low accuracy, Tsinganos et al. [62] employed a threshold-based algorithm with a k-nearest neighbor (kNN) classifier to improve accuracy. Moreover, Lee Y. Y. et al. [52] studied the change in acceleration sensor value and found that the acceleration sensor value’s signal vector magnitude value variation showed a significant difference in daily activities such as walking, running, sitting, and falling. Similarly, Shahzad et al. [59] presented FallDroid, a two-step algorithm that combines the threshold-based method (TBM) and multiple kernels learning support vector machine (MKL-SVM) to effectively identify fall-like events and reduce false alarms. Furthermore, Dogan et al. [43] gathered information from ten users to evaluate their proposed fall detection method, and tested five machine learning classifiers to find effective threshold values. Finally, Harari et al. [46] presented a proof-of-concept fall detection system implemented in a common smartphone to detect real-life falls in real time. This research, combined with that of the aforementioned studies, demonstrates significant progress in fall detection, and provides unique insights for future fall prevention, detection, and treatment.

## 4. Discussion

### 4.1. Background on Fall Detection and Prevention

The world’s elderly population of 60 years and older reached 251.6 million in 1950, 488 million in 1990, and is expected to reach 1205.3 million in 2025, according to United Nations projections. These figures represent a 144% increase between 1950 and 1990 and a 146% increase between 1990 and 2025 [70]. Even if their health deteriorates, the elderly or people with disabilities prefer to remain in their homes [71]. As a result, the telehealth service has been widely implemented and used to assist individuals (e.g., the elderly or people with disabilities) in living independently at home [72,73,74,75,76]. As the aging and disability issues converge, smart home-based health monitoring has emerged as a critical research area for ubiquitous and embedded system computing.

Fall detection and prevention systems are devices designed to detect and prevent falls in elderly individuals. These systems typically consist of sensors and a monitoring device. The sensors are placed around the home in areas where a fall is more likely to occur, such as bathrooms, stairs, and hallways. The sensors detect changes in movement, such as a sudden stop or a decrease in pressure, which may indicate a fall. The monitoring device is then triggered, and an alert is sent to a caregiver or family member. This alert allows them to provide help and assistance if needed.

The primary benefit of fall detection and prevention systems is increased safety and security for elderly individuals. These systems allow elderly individuals to remain independent and safe in their own homes, reducing the risk of injury from falls. In addition, these systems can provide peace of mind to family members and caregivers, knowing that their loved ones are being monitored and that help is available if needed.

Since fragile old persons have trouble walking and are at significant risk of falling, many devices have been developed to detect falls [77]. These technologies fall into two categories. Systems based on sensors and portable devices are (1) nonportable systems and (2) systems [78]. The monitoring area’s environmental sensors, typically artificial vision systems based on cameras [79] or floor sensor systems (pressure, vibration, capacitive, etc.) [80], are used by nonportable systems. Systems based on cameras and fixed sensors do not protect outside of the observation area, necessitating the installation of a costly and complex network of cameras and sensors to protect the home. Users also experience an adverse reaction to feeling watched or monitored by these systems.

Motion sensors, such as accelerometers and gyroscopes, are typically the foundation of portable systems. Due to its portability, affordability, and ability to give movement-related information, accelerometry is a viable choice [77]. Its key drawbacks include battery life limitations, limited processing power, and the necessity for practical algorithms to identify falls in situations where motion artifacts can have the same intensity as falls themselves, leading to many false positives [81]. Most commercial solutions use accelerometers on bracelets or pendants because they are more sensitive to motion artifacts when it comes to positioning portable sensors.

### 4.2. Smartphone Potential for Fall Detection and Prevention

In recent years, SP has been used to send patients reminders, track disease symptoms, promote physical activity and healthy eating, and address various other health issues [82,83,84].

As of 2017, 31% of people aged 75 and older and 49% of people aged 70 to 74 own SPs, respectively [85]. SPs offer a great deal of promise to be used as a tool for balance screening outside of laboratories, because older persons are more likely than younger adults to own an SP and because they are also portable and cost-effective.

Body-worn accelerometers can also be used to assess balance and gait. A systematic review investigated inertial sensors’ use to assess fall risk in older adults [86]. While body-worn accelerometers have the potential to provide accurate and objective fall risk assessment, almost all inertial sensors used in the investigation require personnel assistance to operate, analyze, and interpret the data. SPs may overcome this limitation by taking advantage of the public’s familiarity with SP technology and embedded accelerometers to measure balance objectively. An objective balance assessment that requires little assistance and is simple to use may increase older adults’ acceptance and use of technology. According to a recent review on mobile technology to assess balance and fall risk, while SP technology is becoming a promising tool to evaluate posture, few studies have validated SPs using standard gold techniques [86].

SPs have accelerometers built in that can be used to assess equilibrium. Numerous sensors are integrated into the newest generation of mobile phones. Today’s universal accelerometer sensors included in SPs and mobile phones have the same accuracy qualities as accelerometers designed for a particular application. Moreover, the potent processing power found in SPs allows for the execution of algorithms and other computational activities. Additionally, data can be transferred from SP to other computing devices via wireless technologies such as Bluetooth or WiFi, which is helpful if several sensors are linked to SP to detect or prevent falls. The SP screen enables a simpler and more natural engagement, which is another advantage of SPs over conventional sensors. It is assumed that the inclusion of straightforward, approachable methods of concentration makes it simpler to use these devices in healthcare settings because additional knowledge is not required. The main benefit of the suggested solution is that SPs are widely accessible to most people, and users are less likely to forget to wear SPs than other specialized microsensors.

### 4.3. Sensor Positioning

Despite the promising performance of the SP as a fall detection system reported by multiple studies, it is not widely implemented due to certain limitations. To make SPs a viable solution, two issues must be looked at. First, it is necessary to figure out how to make people wear their smartphones on the waist daily instead of having them in their pockets. To resolve this, external devices such as smartwatches may be used. Several studies utilizing these external sensors have displayed outstanding results.

It is unclear if the system can detect falls if the phone is not mounted on the waist. To address this issue, Aguiar et al. [38] studied whether the accuracy of the system would be different if the phone was stored in the most common places such as the belt level or pocket. The results showed that there were no significant differences in accuracy, with a performance of 97%. To better understand how the system functions in real-life situations, more research is necessary to test its capability of detecting falls while people are engaging in activities such as talking on the phone, texting, putting the SP in their shirt pocket, or trussers pocket.

### 4.4. Methods to Determine a Threshold for Fall Detection and Prevention

Different techniques can be employed to set a threshold for fall detection systems. For example, accelerometer data can be used to obtain a detailed understanding of a person’s movements, allowing for the detection of changes in motion that may signify a fall [25,26,27,28,29,30,31,32,33,34,35,36,37,38,39,40,41,42,43,44,45,46,47,48,49,50,51,52,53,54,55,56,57,58,59,60,61,62,63,64,65,66,67,68]. Machine learning algorithms are a type of artificial intelligence that can analyze data to detect patterns and make predictions [32,39]. Machine learning algorithms can also be utilized to establish a threshold based on the analysis of a dataset containing both falls and nonfalls. They can also be used to predict the likelihood of a fall. This could be used to provide an early warning system for elderly individuals who are at greater risk of falling. By predicting when a fall might occur, medical personnel can take preventative measures to reduce the chances of injury.

Furthermore, a threshold for future falls can be established by studying data from previous falls [25,26,27,28,29,30,31,32,33,34,35,36,37,38,39,40,41,42,43,44,45,46,47,48,49,50,51,52,53,54,55,56,57,58,59,60,61,62,63,64,65,66,67,68]. Finally, user input can be used to develop a threshold for detecting future falls. Many of the studies included in this study used accelerometer data analysis, machine learning, or data from previous falls to establish a threshold and improve detection accuracy [25,26,27,28,29,30,31,32,33,34,35,36,37,38,39,40,41,42,43,44,45,46,47,48,49,50,51,52,53,54,55,56,57,58,59,60,61,62,63,64,65,66,67,68]. Some studies used user input to reduce false-positive fall events, usually by asking the user if they had fallen on their cell phone screen [28].

Because specified thresholds can potentially reduce the number of false-positive fall detection situations, we choose to investigate their application. The false alarms caused by actions such as dropping the phone or a modest fall in which the user is unharmed could be reduced, for example, by applications that use both acceleration and location data to determine a fall. The false alarms are further reduced by allowing the amplitude’s upper threshold to be adjustable. Further processing and comparison with a point are performed on the data produced from the acceleration. A fall is suspected if the processed value exceeds a predetermined threshold. Personal data such as age, weight, height, and degree of activity have also been considered when solving the problem.

### 4.5. Future Applications

Smartphones can be used to detect and prevent falls in the future by utilizing advanced sensors and artificial intelligence technology. Smartphones can be equipped with sensors to detect changes in balance and posture, enabling them to alert the user when a potential fall is imminent. Through the implementation of advanced motion sensing capabilities and applications, smartphones can be utilized to provide reminders to take medications, check in with family and friends, or complete physical therapy exercises. Healthcare professionals can then utilize these data to monitor a person’s health and intervene if necessary.

With the help of artificial intelligence, the smartphone can analyze the user’s gait and offer a variety of corrective actions, such as suggesting a slower walking speed or a more stable posture. AI can also be used to identify hazardous surfaces or obstacles and offer suggestions for avoiding them. Smartphones can also be used to provide reminders to take regular breaks, as well as to alert family and friends if a fall does occur. In addition, smartwatches can be used to provide additional fall prevention by tracking physical activity and offering health tips. The use of smartphones to detect physical changes in a person’s gait, balance, and posture holds the potential to reduce the risk of falls and improve the quality of life for the elderly and disabled. Smartphones have the potential to become an integral part of fall prevention strategies, and the future applications of these devices in fall detection and prevention are abundant.

### 4.6. Limitations

This systematic review has certain limitations associated with it. To begin with, the fact that only studies published in English were included can introduce bias. Moreover, due to the lack of information reported in some cases, it becomes difficult to accurately describe the methods of selecting thresholds and performance. Additionally, there is a risk of misinterpretation of the data and results, which can lead to new variables to consider.

## 5. Conclusions

Fall detection and prevention systems are becoming increasingly important for elderly individuals, allowing them to remain independent and secure in their homes. An upsurge of technology-based applications has been developed to protect at-risk individuals from falls, with the objective of providing more effective and efficient interventions than traditional means. By utilizing convenient and functional fall prevention technology, primarily in the home, elderly people can stay independent while engaging in interventions monitored remotely by healthcare professionals. Successful healthcare technology implementation requires close cooperation between engineers, clinicians, and administrators to understand why falls occur and what interventions could potentially alter the outcome.

## Figures and Tables

**Figure 1 sensors-23-01323-f001:**
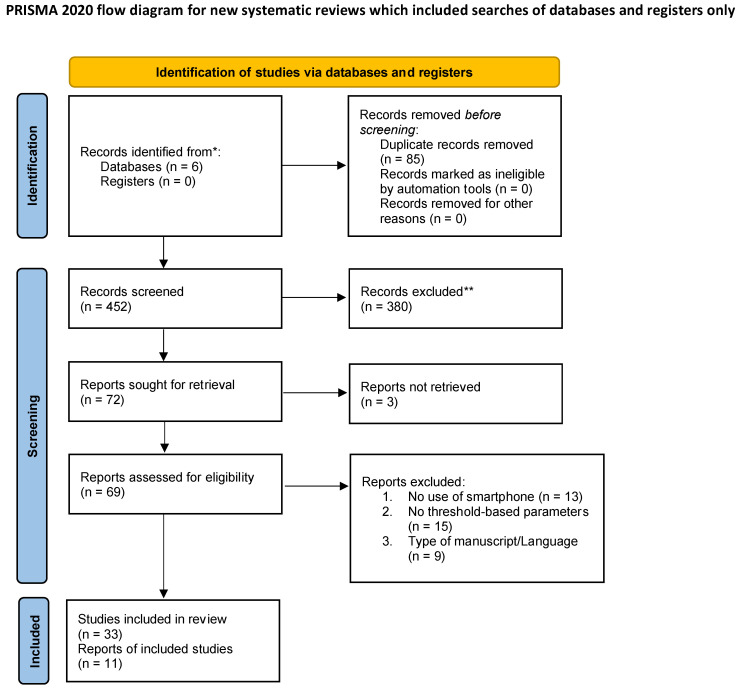
Study selection flow chart. Flow chart describing the study selection process according to PRISMA guidelines. (From: Page MJ, McKenzie JE, Bossuyt PM, Boutron I, Hoffmann TC, Mulrow CD, et al. The PRISMA 2020 statement: an updated guideline for reporting systematic reviews. BMJ 2021;372:n71. doi: 10.1136/bmj.n71; For more information, visit: http://www.prisma-statement.org/) (accessed on 30 August 2022). * Consider, if feasible to do so, reporting the number of records identified from each database or register searched (rather than the total number across all databases/registers). ** If automation tools were used, indicate how many records were excluded by a human and how many were excluded by automation tools.

**Figure 2 sensors-23-01323-f002:**
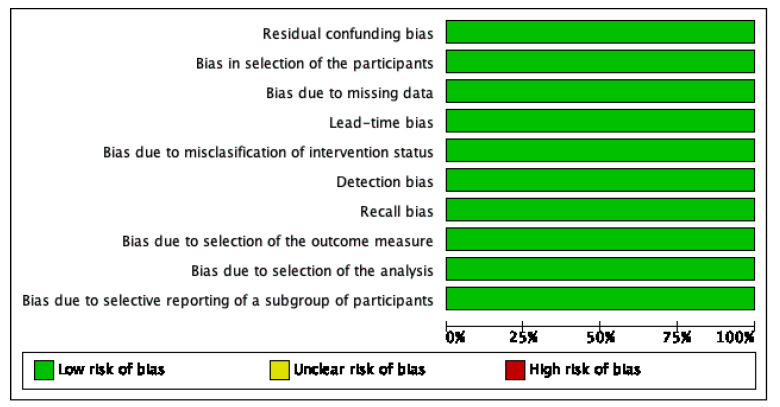
Residual confounding bias: review authors’ judgments about each risk of bias item presented as percentages across all included studies.

**Figure 3 sensors-23-01323-f003:**
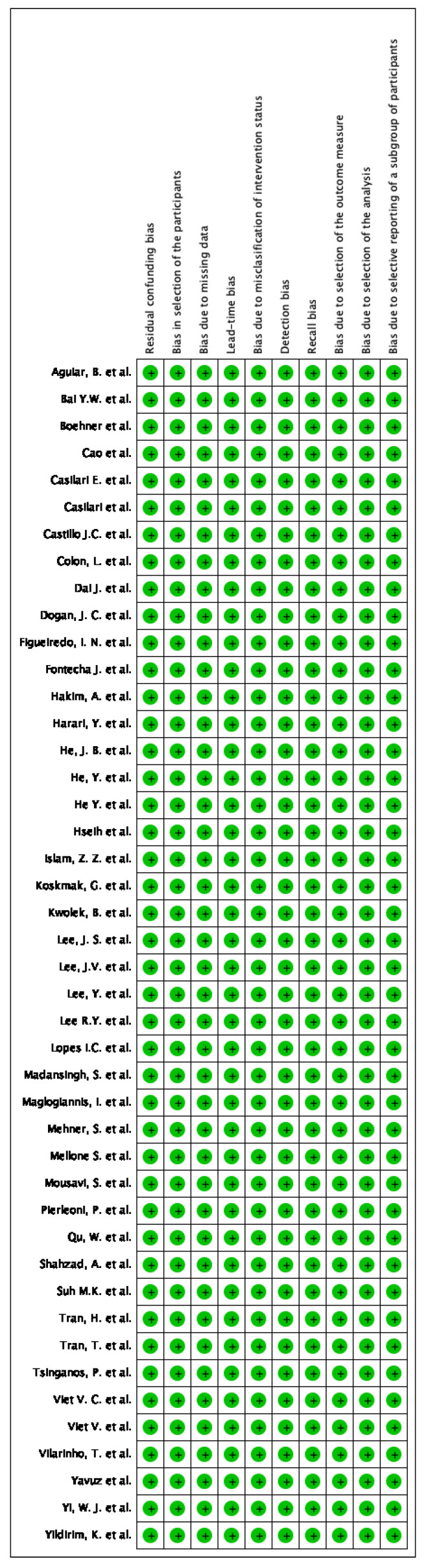
Risk of bias summary: review authors’ judgments about each risk of bias item for each included study.

**Figure 4 sensors-23-01323-f004:**
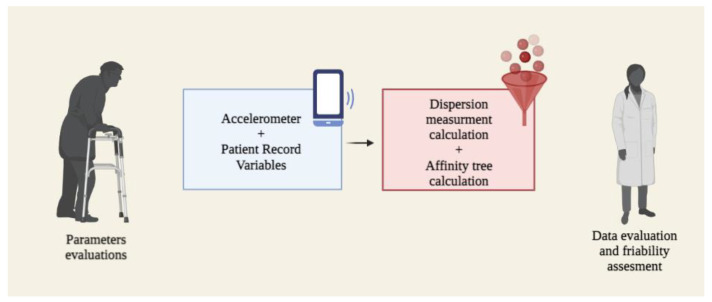
Activity monitoring. Embedded sensors in the SPs can detect a fall. SPs translate the information from sensors and trigger an alarm to caregivers and emergency services based on the user’s capability to give feedback to the SP.

**Figure 5 sensors-23-01323-f005:**
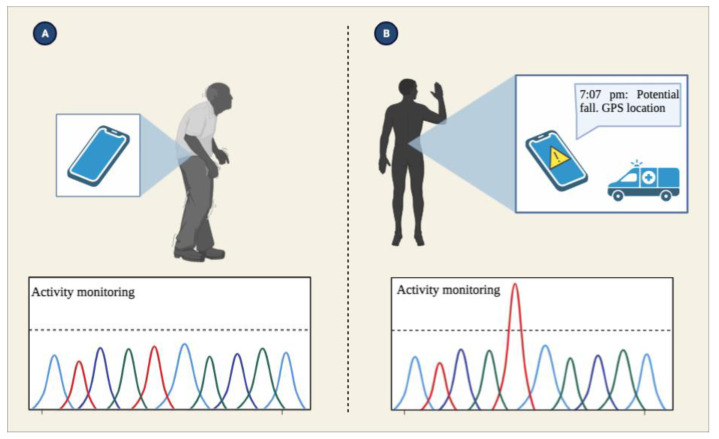
Data processing. Based on specific parameters and with the help of algorithms fed by the SP sensors, researchers can predict the risk of a patient falling. (**A**) Sensors in SP, such as accelerometers, gyroscopes, and magnometers, monitor the patient’s daily activities. (**B**) Once a measurement exceeds the threshold previously stablished, the SP collects and processes sensors data before responding to a potential fall event by alerting emergency personnel and caregivers.

**Table 1 sensors-23-01323-t001:** Description of studies that aimed to detect accidental falls. Abbreviations: EMS: emergency medical services; SP: smartphone; k-nn: k-nearest neighbors algorithm; SVM: support vector machine.

Reference and Year	Smartphone Operative System	Device	Detection (D)/Prevention (P)	Consequence Triggered by the Fall	Parameter	Performance
Dai J. et al. [25] 2010	Android OS	G1	D and P	Speaker sound alert and alarm to guardian	Triaxis accelerometer, gyroscope, and magnetometer	Average false negative: 2.13% Average false positive value is 7.7%
Fontecha J. et al. [26] 2013	Android OS	Not stated	P	Not stated	Triaxis accelerometer	Not stated
Mellone S. et al. [27] 2012	Android OS	Samsung Galaxy SII (GT-I9100)	D and P	Alarm to guardian with GPS location	Accelerometer, gyroscope, and magnetometer	Not stated
Bai, Y-W et al. [28] 2012	Not stated	Not stated	D	Alarm to guardian with GPS location and draw help path	Triaxis accelerometer	Not stated
Castillo J.C. et al. [29] 2014	Android OS	SP (not stated) + external device	D	Alarm to guardian with GPS location	Triaxis accelerometer	Sensitivity: 92.7% Accuracy: 97.2% F-score: 94.8%
He, Y. et al. [30] 2012	Android OS	Lenovo Le-phone	D	Alarm to guardian with GPS location	Triaxis accelerometer	Not stated
Hsieh, K.L. et al. [31] 2019	Not stated	Not stated	D	Not stated	Accelerometer	Not stated
Kwolek, B. et al. [32] 2015	Android OS	SP (not stated) + external device	D	Not stated	Accelerometer	k-nn + acceleration % Sensitivity: 100% Specificity: 92.86% Accuracy: 95.83% Precision: 90.91% SVM + acceleration % Sensitivity: 100% Specificity: 92.86% Accuracy: 91.67% Precision: 83.33%
Lee, J.V. et al. [33] 2013	Android OS	HTC Desire A8181	D	Alarm to guardian	Triaxis accelerometer	Not stated
Lee, R.Y.et al [34] 2011	Android OS	Google G1	D	Speaker sound alert and alarm to guardian with GPS location	Triaxis accelerometer	SP: Sensitivity: 81% Specificity: 77% External accelerometer: Sensitivity: 82% Specificity: 96%
Lopes, I.C. et al. [35] 2011	Not stated	Not stated	D	Speaker sound alert and alarm to guardian with GPS location	Triaxis accelerometer	Not stated
Suh M.K. et al. [36] 2011	iOS and Android	iPhone and Motorola Droid	D	Alarm to guardian	Triaxis accelerometer	Not stated
Aguiar, B. et al. [37] 2014	Android OS	Samsung Galaxy Nexus	D	Alarm to guardian with GPS location	Triaxis accelerometer and biaxial gyroscope	Belt usage Sensitivity: 97.0% Specificity: 98.4% Accuracy: 97.6% Pocket usage Sensitivity: 96.6% Specificity: 98.6% Accuracy: 97.5%
Boehner et al. [38] 2013	Not stated	EZ430 Chronos Texas Instruments Smartwatch	D	Alarm to guardian and EMS	Triaxis accelerometer	Not stated
Cao et al. [39] 2012	Android v.2.2 OS	HTC A3366	D	Alarm to guardian	Accelerometer	Classical algorithm: Sensitivity: 86.7% Specificity: 85.5% Adaptive algorithm: Sensitivity: 86.7% Specificity: 85.5%
Casilari, E. et al. [40] 2016	Android OS	SP and external sensors	D	Alarm to guardian	3-axis gyroscope, 3-axis accelerometer, and magnometer	Not stated
Casilari, E. et al. [41] 2015	Android OS	LG Nexus 5	D	Alarm to guardian	Triaxis accelerometer and gyroscope	Sensitivity: 89.6% Specificity: 95.8%
Colon L. et al. [42] 2014	Android v.4.4.2 OS	Google Nexus 5	D	Alarm to guardian	Triaxis accelerometer and biaxial gyroscope	Precision: 58.2% Specificity: 79% Accuracy: 81.3% Recall: 89%
Dogan, J. C. et al. [43] 2019	Android OS	LG Nexus 5	D	Not stated	Triaxis accelerometer	Accuracy: 95.65%
Figueiredo, I. et al. [44] 2016	Android v.4.1.2 OS	Samsung Galaxy Nexus and Samsung Galaxy Nexus S	D	Alarm to guardian	Triaxis accelerometer	Sensitivity: 100% Specificity: 92.65%
Hakim, A. et al. [45] 2017	Android OS	Sony C6002 Xperia Z	D	Not stated	Triaxis accelerometer	Accuracy: >90%
Harari, Y. et al. [46] 2021	Android v.6.0.1 OS	Samsung Galaxy S5	D	Alarm to guardian	Triaxis accelerometer and gyroscope	Sensitivity: 73% Specificity: >99.9% Accuracy: 97.81%
He, J. et al. [47] 2017	Android OS	Not stated	D	Alarm to guardian with GPS location	Triaxis accelerometer and gyroscope	Sensitivity: 99% Specificity: 95% Accuracy: 95.67%
He, Y. et al. [48] 2012	Android OS	Lenovo Le-phone	D	Alarm to guardian with GPS location	Triaxis accelerometer	Not stated
Islam, Z. Z. et al. [49] 2017	Not stated	Not stated	D	Alarm to guardian	Triaxis accelerometer	Accuracy: >90%
Koshmak et al. [50] 2013	Android OS	Not stated	D	Alarm to guardian with GPS location	Triaxis accelerometer	Senitivity: 90% Specificity: 100% Accuracy: 94%
Lee, J. S. et al. [51] 2019	Android OS	Not stated	D	Alarm to guardian	Triaxis accelerometer	Accuracy: 99.38% Detection rates: 96%
Lee, Y. et al. [52] 2018	Not stated	Not stated	D	Not stated	Triaxis accelerometer	Not stated
Madansingh, S. et al. [53] 2015	iOS	iPhone 4	D	Not stated	Accelerometer, gyroscope, and magnetometer	Not stated
Maglogiannis et al. [54] 2014	Android OS	Pebble smartwatch	D	Alarm to guardian	Triaxis accelerometer	Not stated
Mehner et al. [55] 2013	Android OS	Samsung Galaxy S and Sony Xperia ray	D	Alarm to guardian	Triaxis accelerometer	Detection rate: 83.33% Specificity: 100%
Mousavi, S. A. et al. [56] 2021	iOS v.12.0.1	iPhone 7+	D	Alarm to guardian with GPS location	Triaxis accelerometer	Accuracy: 96.33%
Pierleoni, P. et al. [57] 2015	Android v.4.4.4 OS	Motorola Moto G	D	Alarm to guardian	Triaxis accelerometer and magnetometer	Sensitivity: 99.3% Specificity: 96% Accuracy: 97.7%
Qu, W. et al. [58] 2016	Android v.4.4.3 OS	LG Nexus 4	D	Alarm to guardian with GPS location through social media	Triaxis accelerometer	Not sated
Shahzad, A. et al. [59] 2018	Android v.4.4.2 OS	LG G3	D	Alarm to guardian	Triaxis accelerometer and gyroscope	Sensitivity: 99.52% Specificity: 95.19% Accuracy: 97.81%
Tran, H. et al. [60] 2017	Android v.5.0 OS	Sony Xperia C4	D	Alarm to guardian with GPS location	Triaxis accelerometer	Sensitivity: 60.46% Specificity: 94.80% Accuracy: 82.50%
Tran, T. D. et al. [61] 2016	Android OS	ASUS Zenfone 2	D	Not stated	Triaxis accelerometer	Sensitivity: 93%
Tsinganos, P. et al. [62] 2017	Android OS	LG D160 and ASUS Zenfone 2	D	Not stated	Triaxis accelerometer	Sensitivity: 97.53% Specificity: 94.89%
Viet V. et al. [63] 2011	Android OS	Google Nexus One	D	Not implemented	Accelerometer	Accuracy per category studied: C1: 75% C2: 87.5% C3: 77.9% C4: 84.2%
Viet V. Q. et al. [64] 2012	Android OS	Google Nexus One	D	Not implemented	Accelerometer and orientation sensor.	Sensitivity: 80% Specificity: 96.2% Accuracy: 85%
Vilarinho, T. et al. [65] 2015	Android OS	Samsung Galaxy S3 and Wear Smartwatch LG G Watch R	D	Alarm to guardian	9-axis motion sensor combining a 3-axis gyroscope, 3-axis accelerometer, and 3-axis compass	Sensitivity: 63% Specificity: 78% Accuracy: 68%
Yavuz et al. [66] 2010	Android v2.0 OS	Google Nexus One	D	Alarm to guardian with GPS location	Accelerometer	Meyer wavelet can distinguish falls from nonfalls with an 85% recall while retaining 95% precision
Yi, W. J.et al. [67] 2014	Android OS	Not stated	D	Alarm to guardian	External triaxial accelerometer	Not stated
Yildirim, K. et al. [68] 2016	Android v.2.2 OS	Samsung Galaxy SIII mini	D	Alarm to guardian	Triaxis accelerometer	Not stated

## Data Availability

Not applicable.

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
