# Peer review of "Smartphones and Threshold-Based Monitoring Methods Effectively Detect Falls Remotely: A Systematic Review"

_sensors, 2023, doi:10.3390/s23031323_

Round 1

Reviewer 1 Report (Previous Reviewer 1)

Reviewers' comments have bee addressed. The paper mainly reviews studies of all detection.  However, the paper mentions fall prevention from time to time. It may be better to discussion fall detection and fall prevention in different (distinct) sections in the manuscript.

Author Response

Response to Reviewer 1 Comments

Point 1: Reviewers' comments have been addressed. The paper mainly reviews studies of all detection.  However, the paper mentions fall prevention from time to time. It may be better to discussion fall detection and fall prevention in different (distinct) sections in the manuscript.

Response 1: Thank you for your insightful observation. We also considered this option, but because there aren't that many manuscripts aimed at preventing falls, we believe that having one discussion about fall prevention and detection would be the best option. Furthermore, we notice that these two events are closely related because studies on fall prevention had to develop a system to detect falls in the first place.

Reviewer 2 Report (Previous Reviewer 3)

The changes made to the paper have improved its overall quality and clarified some issues. Therefore, I'm able to suggest its publication.

Author Response

Response to Reviewer 2 Comments

Point 1: The changes made to the paper have improved its overall quality and clarified some issues. Therefore, I'm able to suggest its publication.

Response 1: Thank you for your comments.

Reviewer 3 Report (New Reviewer)

I thank the authors for submitting their work in this journal. Overall the paper is well written and the content is interesting to pursue further interest towards publication. However, I have some concerns and suggestions regarding the paper which I have mentioned below. I recommend that the authors address the comments before it can be considered for publication. 

1.       Moreover, fallers fall again in line 39 can be removed.

2.       Line 41, ”These injuries secondary… ” is not clear. What kind of injuries? Maybe the author can mention one or two such injuries which are secondary to fall.

3.       In search criteria, the authors should also include terms like ”fall detection”. Also, I think the authors should also consider systems where integrated systems including smart phone was used for fall detection.  For example, consider the following paper:

Greene, S., Thapliyal, H., & Carpenter, D. (2016, December). IoT-based fall detection for smart home environments. In 2016 IEEE international symposium on nanoelectronic and information systems (iNIS) (pp. 23-28). IEEE.

4.       Figure 2, it should be residual confounding bias.

5.       The structure of the paper needs to be revisited. Right now, the authors have categorized the studies based on fall prevention studies and fall detection studies. In most cases, falls need to be detected first in order to prevent it. Moreover, in the section where the author discusses fall prevention, studies implementing fall detection system are also discussed. Hence, it would be better if the authors structures the discussion of the studies in a different way. Maybe based on the type of sensing modalities used or platform used. 

Author Response

Response to Reviewer 3 Comments

Point 1: Moreover, fallers fall again in line 39 can be removed.

Response 1: Thank you for your observation. We have removed that sentence from the manuscript.

Point 2: Line 41, ”These injuries secondary… ” is not clear. What kind of injuries? Maybe the author can mention one or two such injuries which are secondary to fall.

Response 2: Thank you for your observation. We have modified the sentence to: “non-fatal injuries secondary to falls are one of the top 20 most expensive conditions, with a medical cost of $51 billion in 2015 that is expected to rise to $100 billion by 2030.” We consider this sentence is more according to the reference cited.

Point 3: In search criteria, the authors should also include terms like ”fall detection”. Also, I think the authors should also consider systems where integrated systems including smart phone was used for fall detection.  For example, consider the following paper:

Greene, S., Thapliyal, H., & Carpenter, D. (2016, December). IoT-based fall detection for smart home environments. In 2016 IEEE international symposium on nanoelectronic and information systems (iNIS) (pp. 23-28). IEEE.

Response 3: Thank you for your comments. We have included “fall detection” in our MeSH term search. Many of the manuscripts included do not disclose the type of system used because they are conference articles with length constraints or because their manuscripts focus on the type of artificial intelligence and algorithms used. The scope of our manuscripts was to describe fall detection/prevention systems using smartphones.

Point 4: Figure 2, it should be residual confounding bias.

Response 4: Thank you for your comments. We have modified the legend of figure 2 for: Residual confounding bias”.

Point 5: The structure of the paper needs to be revisited. Right now, the authors have categorized the studies based on fall prevention studies and fall detection studies. In most cases, falls need to be detected first in order to prevent it. Moreover, in the section where the author discusses fall prevention, studies implementing fall detection system are also discussed. Hence, it would be better if the authors structures the discussion of the studies in a different way. Maybe based on the type of sensing modalities used or platform used. 

Response 5: Thank you for your feedback. Our research aimed to inform readers about the current uses of smartphones for fall detection/prevention. One of our study's limitations was the scarcity of data on fall prevention. Given the close relationship you mentioned, we decided to combine fall detection and fall prevention. Furthermore, after reviewing the included studies, we discovered that all the manuscripts reported using accelerometers and gyroscopes to detect falls because they have higher accuracy, sensitivity, and specificity than other types of sensing modalities, such as cameras, pressure sensors, or microphones. Given the manuscript's length and scope, we will continue our research with a manuscript devoted solely to analyzing the platforms used by the systems to determine a fall event.

Round 2

Reviewer 3 Report (New Reviewer)

I thank the authors for their revision. My comments are satisfactorily addressed and I hence recommend the manuscript for publication.

This manuscript is a resubmission of an earlier submission. The following is a list of the peer review reports and author responses from that submission.

Round 1

Reviewer 1 Report

Only 12 studies were included in the systematic review.  More studies should be included for meaningful coverage.

Since the focus is threshold-based fall detection, it is important to compare and discuss the sensitivity and specificity figures of the studies.  These details are only given in 2 out of the 12 studies.

Reviewer 2 Report

This manuscript is a review of the smartphone usage in fall detection and prevention. 12 studies were included and simply analyzed. I think this work would provide helpful information for the research and development of fall detection and prevention technologies. But I have some questions and suggestions.

1. What is the definition of “fall prevention” in this review? From Table I, I cannot get the differences of the reports with or without prevention.

2. Some details needs to be discussed to increase the value of this review. For example, how to decide the thresholds in the mentioned reports, is there any performance variation between different methods for threshold determination, and what is the reason of variation?

3. Some existing bottlenecks, technical trends and/or future applications can be summarized in the review work. This would help the researchers of this area a lot.

Reviewer 3 Report

The authors present a sounding work entitled "Smartphones and Threshold-Based Monitoring Methods Effectively Detect Falls Remotely: A Systematic Review.". Unfortunately, the following concerns do not allow me to recommend the publication of this work.

Major concerns:

  • 7 of the 12 papers included in this review are ten years old or more!
  • The authors highlight the importance of fall detection, but unfortunately, only two of the included papers aimed to identify falls and prevent them.